# Sleep Health Promotion Interventions and Their Effectiveness: An Umbrella Review

**DOI:** 10.3390/ijerph18115533

**Published:** 2021-05-21

**Authors:** Uthman Albakri, Elizabeth Drotos, Ree Meertens

**Affiliations:** 1Department of Health Promotion, Care and Public Health Research Institute (CAPHRI), NUTRIM School of Nutrition and Translational Research in Metabolism, Maastricht University, 6200 MD Maastricht, The Netherlands; e.drotos@maastrichtuniversity.nl (E.D.); r.meertens@maastrichtuniversity.nl (R.M.); 2Department of Public Health, Faculty of Applied Medical Sciences, Albaha University, Albaha 65779, Saudi Arabia

**Keywords:** sleep, sleep hygiene, effectiveness, program evaluation, public health, systematic review

## Abstract

Sleep is receiving increasing attention in public health. The aim of this umbrella review is to determine what non-pharmacological sleep health interventions have been evaluated among healthy populations, by examining target groups, settings, and effectiveness in improving sleep quality and duration. Comprehensive searches were conducted in five electronic databases (January 1975–February 2019), yielding 6505 records. Thirty-five articles were selected meeting the following eligibility criteria: (1) systematic reviews or meta-analyses of (2) sleep health interventions in (3) primarily healthy populations. Two reviewers independently screened for inclusion, extracted the data, and assessed the review quality. This umbrella review was registered with PROSPERO (CRD42019126291). Eleven intervention types were defined, and their effectiveness discussed. Substantial evidence demonstrated the effectiveness of later school start times, behavior change methods, and mind–body exercise. Other intervention types, including sleep education or relaxation techniques, demonstrated some promising impacts on sleep, but with less consistent evidence. Results were limited by high heterogeneity between studies, mixed results, and variable review quality. Nevertheless, this umbrella review is a first step towards understanding the current state of sleep health promotion and gives an overview of interventions across the lifespan.

## 1. Introduction

Sleep health has been defined as “a multidimensional pattern of sleep-wakefulness, adapted to individual, social, and environmental demands, promoting physical, and mental well-being” [1]. Insufficient sleep may contribute to chronic diseases, such as obesity [2], cardiovascular disease [3], and diabetes [4]. Lack of sleep may also lead to depression [5], other mood disorders [6], and reductions in cognitive performance, including memory and learning difficulties [7]. Workplace injuries, accidents, and medical errors can also result from insufficient sleep, as daytime drowsiness and fatigue can diminish alertness and decrease reaction time [8,9].

Poor sleep health is a global issue, and studies show an increasing prevalence of inadequate sleep [10,11]. One study estimated that, by 2030, the total number of older adults with sleep problems in low-income countries will be 260 million, an increase from 150 million in 2010 [10]. Ohayon (2011) estimated that the prevalence of people with sleep deficits in various countries ranged from 20% to 41.7% [12]. Worldwide, insufficient sleep affects every age group, although in many countries it remains unidentified and underreported [10,13]. Poor sleep also negatively affects the world economy. A combined estimate for the U.S., Canada, the UK, Germany, and Japan, put the annual economic loss due to sleep problems at USD 680 billion [14].

Most large-scale public health education programs and campaigns have been created to influence diet and exercise, without considering sleep [15,16]. Likewise, the US health agenda in 2010 included guidelines for physical activity and diet as important health-related behaviors, and sleep was not included [17]. However, sleep has recently made its way onto the US health agenda [18], and sleep deprivation reduction is also an emerging public health priority in the UK [19].

Therefore, the aim of promoting healthy sleep is expected to receive growing attention in the next decades. Sleep health promotion involves improving sleep duration and quality. Such work targets sleep-related health behaviors and knowledge among healthcare professionals, policy makers, and the general population [20,21]. Sleep duration is normally defined as the cumulative amount of sleep during the nightly episode of rest, or over a 24-h period [22]. Sleep quality is described as “one’s satisfaction with the sleep experience, integrating aspects of sleep initiation, sleep maintenance, sleep quantity, and refreshment upon awakening” [23]. Measures and tools for measuring sleep duration and quality vary, and can be classified as objective or subjective.

Several systematic reviews and meta-analyses have been conducted on sleep health interventions, examining different intervention types, target groups, and settings. However, an ‘umbrella review’, providing an overview of all systematic reviews published across the wide-ranging field of sleep health promotion, is missing from the literature. The purpose of this umbrella review is to systematically summarize the scope and effectiveness of sleep interventions in primarily healthy populations, i.e., not diagnosed with a sleep disorder or specific disease. More specifically, it aims to determine what kinds of non-pharmacological sleep health interventions have been implemented, in what target groups and settings, and how effective they are in improving sleep quality and duration. This overview of sleep health interventions across the lifespan is also meant to introduce interested researchers to this relatively new field of sleep health.

## 2. Materials and Methods

### 2.1. Search Strategy and Selection Criteria

The Preferred Reporting Items for Systematic reviews and Meta-Analyses (PRISMA) guidelines were adapted for conducting this umbrella review [24]. A systematic search was implemented using six online bibliographic databases/search engines: PubMed, Medical Literature Analysis and Retrieval System Online (MEDLINE), the Psychological Information Database (PsycINFO), Excerpta Medica dataBASE (Embase), the Cumulative Index to Nursing and Allied Health Literature (CINAHL), and the Cochrane Library, from 1975 to January 2019. The search terms “health education”, “intervention”, “sleep”, “program evaluation”, and related synonyms, were used in the search strategy, including Medical Subject Headings (MeSH) terms in PubMed and thesaurus terms in PsycINFO. A filter was used to limit the retrieved articles to systematic reviews and meta-analyses. The search syntax is available in Appendix B.

Selection criteria were established using the PICOS strategy (Population, Intervention, Comparison, Outcome, Study design). Articles had to include (1) primarily healthy populations (i.e., systematic review/meta-analysis should not be directed at populations with a specific diagnosed (sleep) disorder); (2) sleep health promotion interventions; (3) any public health intervention settings (i.e., not laboratory settings); (4) the evaluation of sleep duration and/or quality; and (5) a systematic review or meta-analysis study design. Only references published in English were eligible. Reviews involving a majority of research participants diagnosed with sleep disorders or other specific conditions, like cancer, were excluded. Articles were also excluded if they were not published in scientific journals, e.g., theses, book chapters, and conference proceedings. This review was registered a priori with PROSPERO (CRD42019126291).

### 2.2. Screening and Data Extraction

After removing duplicates, two researchers (U.A. and L.P.) first screened the titles and then the abstracts, using the selection criteria. In cases of disagreement, articles were included in the next step. The researchers then performed full-text screening and any disagreements were resolved through discussion with a third reviewer (R.M.).

Two reviewers (U.A. and E.D.) extracted the data independently and any disagreements were resolved through consultation. The Joanna Briggs Institute form was adapted for data extraction. Designs of included studies were categorized into three types: RCTs, crossover study, and quasi-experimental. The quasi-experimental category comprised controlled non-randomized, pre-/posttest, and time series designs [25].

While extracting the data, definitions of the intervention types were developed considering the interventions most commonly found in the literature. The studies of each review were categorized as one of the defined intervention types; this allowed grouping of particular sleep promotion methods when examining their prevalence and effectiveness. Individual studies from selected reviews were “excluded” and not considered in the conclusions if they did not aim to measure any relevant sleep-related outcome, were conducted in a laboratory setting, or had unsuitable study designs for interventions. Results based on single studies were not listed in the results table, to avoid giving unjustified weight to individual studies.

### 2.3. Methodological Quality Assessment

Methodological quality was assessed using the JBI Critical Evaluation Controller for Systematic Reviews, which consists of 10 items, awarded 0 or 1 [26]. Quality scores of 0–5 were considered weak, scores of 6–8 were moderate, and scores of 9 or 10 were strong. Two reviewers (U.A. and E.D.) performed the assessment independently and disagreements were resolved by discussion with a third reviewer (R.M.).

A common limitation in umbrella reviews is the overlap of primary evidence. This occurs when different reviews reference the same component studies, leading to “double-counting” of results and overstating evidence [27]. A “corrected covered area” (CCA) measure, developed by Pieper et al. (2014), was calculated to evaluate this degree of overlap in the primary data [28].

## 3. Results

### 3.1. Description of Reviews

#### 3.1.1. Included Reviews, Studies, and Quality Assessment

Thirty-five systematic reviews were included after screening, as depicted in Figure 1. Seventeen included meta-analyses. The number of studies included in these reviews ranged from 4 [29,30] to 112 [31]. A total of 552 primary studies were included. After examining the overlap between studies reported in multiple reviews, only 443 unique studies were found. Fifty-six studies were found in two reviews, and 21 other studies were included in three or more reviews. The CCA value was calculated at 0.72, which is interpreted as only a small overlap of the component studies [28].

The quality assessment ratings are presented in Appendix A: Ratings of the included reviews using the JBI Quality Assessment Criteria for Systematic Reviews. Twenty-one reviews were rated high quality, 11 moderate, and three low. For most reviews, the question, inclusion criteria, search strategy, sources used, and criteria for appraising studies were stated clearly. However, the assessment of potential publication bias was particularly lacking, and was not assessed in 20 reviews.

#### 3.1.2. Intervention Types

Eleven intervention categories were grouped, mainly on the basis of the techniques used in the interventions and not, for example, on the theoretical background (Table 1). The intervention types were sleep education, behavior change methods (BCM), relaxation techniques, physical exercise, mind–body exercise (MBE), aromatherapy and/or massage, psychotherapy, environmental interventions, and later school start times. Some reviews also included multicomponent interventions that incorporated multiple intervention types. There were also some other, less commonly observed interventions, such as hypnosis, biofeedback, and magnet therapy, which were gathered into an “other” category.

#### 3.1.3. Target Populations

The characteristics of the individual reviews are summarized in Table 2, by age categories. Five reviews included only infants and young children (0–5 years) [29,32,33,34,62], while ten reviews included school-aged children and/or adolescents (5–17 years) [35,36,37,38,39,40,47,59,60,61]. More than half of the reviews examined adults (>17 years), and four focused exclusively on elderly populations [30,31,41,42,43,44,45,46,48,49,50,51,52,53,54,55,56,57,58,63]. Some reviews included only very specific groups, such as college students [30,44], shift workers [41,49], athletes [42], or pregnant women [51].

#### 3.1.4. Intervention Settings

Most reviews included studies that were conducted in multiple settings [29,31,32,33,34,35,36,38,41,43,45,46,49,51,52,57,58,62,63]. In reviews that focused on single intervention settings, these included schools [37,39,59,60,61], universities [30,44], the community [54,55], the workplace [49], outpatient settings [47], hospitals [48,50], nursing homes [56], and gyms [53]. One review did not clearly report the setting [42].

#### 3.1.5. Sleep Measures

Sleep duration was usually measured using the total sleep time (TST) in a single night or the average TST across a period of time. Sleep quality was measured through composite scores on questionnaires, like the frequently used Pittsburgh Sleep Quality Index (PSQI). Sleep onset latency (SOL) was also considered a sleep quality measure. In infants and young children, the number of night awakenings and bedtime problems were considered indicators of sleep quality [34,62].

Both objective and subjective tools were used to measure outcomes. Actigraphy and polysomnography were commonly used objective tools. Subjective tools included many sleep indices, such as the PSQI or the Insomnia Severity Index, as well as other self-report tools like parent reports, sleep diaries, interviews, and non-standard questionnaires. Twenty-nine reviews included a combination of subjective and objective measures. Four reviews included studies that only measured sleep outcomes subjectively, while two reviews did not identify the measuring techniques.

### 3.2. Types of Interventions and Their Effectiveness

Table 2 displays the review characteristics, including types and descriptions of interventions, number of eligible studies, study designs, populations, and the results by review. Below, each intervention type and the evidence of its effectiveness is reviewed systematically.

#### 3.2.1. Sleep Education

Seventeen reviews included sleep education, with wide variations in approaches [29,30,32,33,34,35,36,37,38,39,40,41,42,43,44,45,46]. These interventions included information on sleep health, sleep cycles, consequences of insufficient sleep, and/or sleep hygiene tips. However, some reviews provided only vague descriptions of the content included. Many methods were used to deliver sleep education, including seminars, pamphlets, telephone calls, and online information. Targeted populations for sleep education varied, but college students and parents with infants and young children were common.

Ten out of the seventeen reviews showed small positive effects of sleep education on sleep duration and/or quality measures [29,32,33,34,35,37,38,39,42,43], though in three, the effects were not maintained at follow-up [35,38,39]. For the remaining reviews, two showed no significant effects on sleep duration [44,46], and one reported insufficient evidence [30]. Although sleep education interventions were included in the meta-analyses in four other reviews, they were combined with different intervention types, so no conclusions could be drawn [36,40,41,45]. In two reviews, interventions aimed to increase sleep knowledge in school settings [30,37]. While these reviews showed increases in sleep-related knowledge, this was not accompanied by any change in sleep-related behaviors or sleep quality/duration. While some sleep education interventions demonstrated small positive impacts on sleep duration and/or quality, the results were quite mixed.

#### 3.2.2. Behavior Change Methods (BCM)

Eleven reviews covered BCM using varying techniques for different target populations [29,32,33,34,38,40,42,45,47,48,49]. A common target group for BCM was infants and children, with methods implemented by their parents. Examples of BCM are standardized bedtimes, scheduled awakenings, positive routines, controlled comforting, and gradual extinction (i.e., parents leave children alone for extended periods, ignoring protests and crying) [34,40,47]. Prescribed sleep wake schedules were used in infants, athletes, and shift workers [42,45,49]. BCM were also often employed in combination with sleep education.

In one review, each for infants, athletes, and shift workers, improvements in sleep duration and/or quality were demonstrated [33,42], while one review did not report sleep duration outcome data [49]. Three other reviews with infants and children showed improvement in sleep quality [34,40,47], and another showed improvement in both sleep duration and quality [32]. Two reviews demonstrated inconsistent evidence [38,48], and one did not report specifically on the effectiveness of BCM [45]. Overall, there is substantial evidence that BCM in infants and children (as managed by parents) increases sleep duration and quality [29,32,33,34,40,47]. There is also some encouraging (but more limited) evidence for other populations.

#### 3.2.3. Relaxation Techniques

Nine reviews examined relaxation techniques [31,35,41,44,45,48,50,51,52]. Participants varied in the reviews (adolescents, shift workers, college students, pregnant women, and hospitalized adults). The diversity of techniques used in this category was notable, ranging from progressive muscle relaxation, to mindfulness, relaxing music, etc.

Improvements in sleep duration and/or quality varied from small to large positive effects. Sleep quality was measured in most reviews, while fewer examined sleep duration. One review showed that varying relaxation techniques improved sleep quality with a medium effect size [44]. One review demonstrated that relaxation had small to large positive effects on sleep quality [50], while another review demonstrated mixed results on sleep duration and quality [31]. In five reviews, relaxation techniques were not separately described in terms of effectiveness [35,41,45,48,51]. The strongest evidence in this category was for listening to relaxing music, supported by a meta-analysis [52]. Evidence for other relaxation techniques was more mixed.

#### 3.2.4. Physical Exercise Interventions

Seven reviews assessed the effect of physical exercise on sleep outcomes, including aerobic exercise [45,51,53,55], shadow boxing [54], Pilates [46], and low-intensity exercise [56]. The participant groups varied, but the reviews targeted more women than men, including some specific subgroups: postmenopausal, pregnant, postpartum, and middle-aged women.

The effectiveness of physical exercise in improving sleep also ranged from small to large effects. One review showed small effects on sleep quality [55], while three reviews showed large effects on sleep quality [46,51,53]. Mixed results on sleep duration were reported in one review [56], while another review showed medium effects on sleep duration [54]. One meta-analysis combined a physical exercise study with different interventions, not reporting the effectiveness of physical exercise alone [45]. Physical exercise interventions have shown some promising improvements in sleep duration and quality, but the sample sizes in these reviews were relatively small.

#### 3.2.5. Mind–Body Exercise (MBE)

Mind–body exercise was categorized as a separate intervention type, combining physical activity with meditative components. Six reviews included studies that used MBE [31,51,53,54,55,57]. Common forms of MBE were tai chi, yoga, and Qigong, but there were also less-common techniques, such as the Rességuier method, which promotes patient awareness of bodily perceptions and control. Participants varied, but older adults were often targeted.

The effectiveness of MBE to improve sleep was mixed, but quite consistently positive. One review reported that tai chi had medium effects on sleep duration and large effects on sleep quality [54], while another reported that tai chi had small effects on sleep quality [55]. Another review assessed different types of MBE, and reported positive effects on sleep duration and quality [31]. For yoga interventions, one review reported improvement in sleep quality [51], while another showed no significant effects [53]. Moreover, a meta-analysis pooled data from 14 MBE interventions, demonstrating medium effects in improving sleep quality in older people with sleep complaints undiagnosed with sleep disorders [57]. MBE was investigated in many studies, including several RCTs, and consistently showed positive impacts, particularly in older adults.

#### 3.2.6. Aromatherapy and/or Massage Interventions

Many interventions combined both aromatherapy and massage, so these two techniques were placed into one category. Six reviews examined aromatherapy and/or massage [46,48,50,51,56,58]. Participants varied, and these interventions were often conducted in healthcare settings, such as hospitals and nursing homes.

Two reviews demonstrated that massage combined with aromatherapy has larger effects on sleep quality than aromatherapy alone [46,50]. In contrast, one meta-analysis demonstrated the opposite; while sleep quality consistently improved, a subgroup analysis demonstrated that inhalation aromatherapy was more effective than a combined approach [58]. One review reported massage as the most promising approach to improve sleep quality among pregnant women compared to other interventions [51]. Additional reviews included interventions using both aromatherapy and massage, but the results were inconsistent [48,56]. Overall, there is some evidence of improvement in sleep quality with both massage and aromatherapy.

#### 3.2.7. Environmental Interventions

Five reviews included environmental interventions [46,48,49,50,56]. There was notable variation in the techniques used to modify the environment to promote sleep. For instance, daytime bright-light therapy, through the influence of circadian rhythms, was used to improve sleep duration and quality among shift workers and hospitalized patients [48,49]. Nature sounds, white noise, or noise reduction were different techniques to augment the auditory environment [48,50]. Other interventions included infants staying in their mother’s hospital rooms [46], adjusting bedroom temperature levels, and reducing sleep interruptions for nighttime nursing [56]. Environmental interventions were often conducted in monitored facilities, such as hospitals or care facilities.

Evidence of the effectiveness of environmental interventions is mixed. One review reported that bright-light therapy and nature sounds had a large positive effect on sleep quality [50], and another reported that bright light improved sleep duration and/or quality, but not significantly [49]. Another review concluded that sleep duration and quality were improved through the use of bright light and white noise [48]. However, noise reduction and reducing night-time nursing care did not show increases in sleep duration [56]. One review examined environmental interventions with other interventions, and did not report specific results separately [46]. High heterogeneity and few studies were found in the literature, providing very limited evidence.

#### 3.2.8. Psychotherapy

Four reviews included psychotherapy, conducted online, in groups, or one-on-one with therapists [40,41,44,46]. Most studies employed cognitive behavior therapy (CBT), but other psychotherapeutic interventions were also observed, such as implosive therapy and constructive worry. Participants varied, including college students, shift workers, and postpartum women.

The reported effectiveness of CBT differed. One review examined CBT and other psychotherapeutic methods, and it showed that CBT had large positive effects on sleep duration, SOL, and other sleep quality metrics. In the same review, other psychotherapeutic interventions showed medium impacts on sleep duration and some sleep quality measures, but not SOL [44]. This review provided the most substantial evidence for psychotherapy. The remaining three reviews combined CBT studies with other interventions, making it difficult to determine the effect of psychotherapy [40,41,46]. In the present umbrella review, we found limited evidence for the use of psychotherapy to promote sleep health in healthy populations.

#### 3.2.9. Later School Start Interventions

Four reviews examined this intervention type, and participants were adolescents and children in formal school settings. School start times were delayed 20–85 min. Two meta-analyses reported an increase in sleep duration: one with a MD of 18–65 min and the other with a MD of 1.39 h [59,61]. In an additional review, the values of sleep duration increasing ranged from 25 to 77 min across the included studies [60]. The demonstrated effect substantially increased when start times were delayed more than 60 min in comparison to the controls. Overall, very large sample sizes were included in these interventions. This may suggest strong evidence for later school start times, despite the lower number of reviews.

#### 3.2.10. Multicomponent Interventions

Eighteen reviews included multicomponent interventions, which combined techniques from multiple categories [29,31,32,33,34,36,37,38,40,41,44,45,48,49,50,51,56,62]. Interventions that combined sleep education and BCM targeting infants and children were common and showed positive effects on sleep duration and/or quality [29,32,33,34,62]. The substantial heterogeneity between the multicomponent interventions across different reviews preclude drawing a firm conclusion on their effectiveness.

#### 3.2.11. Other Types of Interventions

Seven reviews included other less commonly observed interventions. These were dietary interventions, hypnosis, biofeedback therapy, magnet therapy, drinking herbal tea, acupuncture, cryostimulation (where the body is exposed temporarily to extremely cold temperatures), and infrared light treatment [31,42,44,46,50,51,63]. One review examined only dietary interventions, which aim to improve sleep by finely manipulating nutritional intake [63]. The review demonstrated no effects in natural (non-laboratory) settings. One review showed that infrared light irradiation and cryostimulation improved sleep duration and/or quality in athletes [42]. Acupuncture was reported in two reviews [50,51], and one demonstrated positive effects on sleep quality [51]. The effectiveness of other interventions varied, and there were few studies.

## 4. Discussion

Sleep health promotion is an emerging issue in public health. The aim of this research was to conduct an umbrella review, a review of reviews, of sleep health interventions in the general population. Common types of interventions, target groups, and settings were identified, and conclusions were drawn regarding effects on sleep duration and quality.

The scope of the reviews varied widely. Some were very broad, including all non-pharmacological sleep interventions and addressing multiple target groups. Reviews targeting only specific groups, such as shift workers, often included a wide range of intervention types, providing overviews of all sleep promotion interventions in that group. Other reviews only included one intervention type and addressed only one target group (e.g., later school times for children/adolescents). Furthermore, the methodological quality also varied, though many reviews were rated as high quality. Lower-quality reviews reported insufficient details on interventions, designs, methods, and results of the included studies, and/or failed to provide concrete recommendations. Only 7 of the 35 reviews adopted the PRISMA guidelines for reporting.

The most commonly observed intervention types were sleep education and behavior change methods. Sleep education interventions consist of providing basic education about sleep (e.g., what is sleep and its health benefits), often combined with sleep hygiene tips (e.g., ‘No caffeine in the evenings’). Sleep education is frequently conducted in school-aged populations, such as children or college students. Behavior change methods are interventions based on behavioral theories, i.e., strategies to improve sleep by augmenting certain associations with sleeping. Examples of this include using bedtime routines for children, or not rewarding attention seeking and crying at bedtime. As the examples show, behavior change methods often target infants, although some adult subgroups, such as athletes and shift workers, were targeted as well [42,49].

Another common intervention was relaxation techniques. This could involve using such techniques during the day (such as mindfulness) or specifically around bedtime (e.g., progressive muscle relaxation, and listening to music). In interventions defined as ‘mind–body interventions’, meditative techniques are combined with physical exercise, such as tai chi. In contrast, physical exercise interventions only involved physical activity to improve sleep, without explicit relaxation components (e.g., aerobics). While these three intervention types were observed in various target groups, mind–body and physical exercise were investigated in adults in particular [31,53,54,55,57].

Fewer reviews included aromatherapy and/or massage, which involved the use of fragrant oils that are inhaled or massaged into the skin. Massage is implemented alone or often in combination with aromatherapy, involving manual techniques implemented by a therapist (e.g., back massage and foot reflexology). These interventions are often observed in healthcare facilities, such as nursing homes. Another less commonly observed sleep intervention type is environmental interventions, the modification of sleep environments. Examples include bright light and noise or temperature adjustment, the techniques most often used in healthcare settings. Later school start times were also less commonly observed, which involved changing the time of school starts to correspond with the circadian rhythms of adolescents (teenagers undergo a delay in their sleep–wake rhythm, as a consequence of biological processes during puberty) [64]. However, this is the most common intervention for adolescents, when considering the number of study participants. School districts may set policies to start school days at 9:00 instead of 8:00, as is recommended by the included reviews [59,60,61]. Likewise, the American Academy of Pediatrics recommends that middle and high schools aim to start no earlier than 8:30 [65].

The most common example among the therapies was cognitive behavioral therapy, which aims to support patients in identifying and changing destructive or disturbing patterns of thoughts that negatively affect behavior and emotions [66]. It should be noted that cognitive behavioral therapy for insomnia is widely acknowledged as an effective treatment for people diagnosed with insomnia [67]. However, in the present review, addressing the general public, evidence for its effectiveness was limited. Lastly, multicomponent interventions were commonly observed. In particular, sleep education was often combined with behavior change methods, though physical exercise, mindfulness, and environmental modifications were sometimes combined with sleep education too.

Although the 11 defined intervention types adequately describe the sleep health interventions in the included reviews, the distinctions between the intervention categories were sometimes not as straightforward as might be assumed. For example, giving sleep hygiene tips was considered sleep education, but these tips often suggest bedtime relaxation or creating a dark, quiet sleeping environment. One could then argue that education on sleep hygiene also has relaxation and environmental components. Furthermore, CBT and BCM partly share their theoretical underpinning. Nevertheless, CBT and BCM were considered as separate intervention types, as techniques as well as implementers varied considerably. However, in determining the 11 intervention types, the main focus of an intervention could always be identified.

Three categories showed substantially more evidence for improving sleep duration and/or quality: behavior change methods, mind–body exercise, and later school start times. These categories consistently demonstrated statistically significant improvements with relatively large effect sizes. Many reviews featuring behavior change methods and mind–body exercise included rigorous RCT study designs. This included two meta-analyses primarily analyzing behavior change methods [33,62], and two meta-analyses of exclusively mind–body exercises [54,57]. Later school start times also demonstrated strong effects in two reviews with particularly large participant samples. The eight additional intervention types defined in this review also demonstrated some promising impacts on sleep, but with less research conducted and/or less consistent evidence. Some techniques demonstrated higher effectiveness than others within categories. For example, listening to music seemed to be more effective than other relaxation techniques.

These conclusions should be interpreted with caution. There was heterogeneity in study designs, outcome measures, populations targeted, and specific techniques. Generalization is another concern, as some data is from narrow target groups or settings and results may not be applicable in other contexts. For instance, physical exercise to improve sleep was mostly implemented for women, and environmental interventions were mainly applied in healthcare settings. These interventions may not demonstrate similar effectiveness in other populations or settings. Moreover, there were contradictory findings regarding the relative effectiveness of some specific techniques within categories. For example, one meta-analysis showed aromatherapy to be more effective than massage [58], while another showed that massage was the more effective technique of the two [46].

### 4.1. Strengths and Limitations

This review had some notable limitations. First, overlapping of primary studies between reviews is a common limitation in umbrella reviews. However, only a small amount of overlap was detected with the CCA calculation [28]. There was a high degree of heterogeneity, which made direct comparisons between reviews impossible. Publication bias may have limited what studies were published and included in reviews, which in turn may be a limitation of this umbrella review. Likewise, the quality of an umbrella review depends on the quality of the included reviews, which in turn depends on the quality of the primary studies. Therefore, there may be some undetected sources of bias. However, it is reassuring that the quality scores of most included reviews were strong.

A strength of this review was the broad variety of electronic databases in its search strategy, allowing a broad overview of all possible types of non-pharmacological interventions used across all age groups and settings. Two reviewers conducted the entire screening, data extraction, and quality assessment independently, increasing the validity of the extracted data and methodological strength of the research.

### 4.2. Implications for Practice and Future Research

Sleep health promotion has been gaining attention in public health, and effective interventions are being developed that improve sleep duration and quality in the general population. Currently, policies regarding sleep have been implemented within different countries and organizations, including later school start times, regulations for shift worker hours, and public education on sleep health [59,68]. Practitioners and policy makers may profit from the insights of this present review to extend such initiatives.

This review also suggests recommendations for future research. As previously mentioned, some intervention types have only targeted specific groups or have not been investigated thoroughly. For instance, mind–body and physical exercise has had promising results among adults and elderly people, so future research could demonstrate if this approach would be effective in children or adolescents. In this age of smartphone technology, apps may be a new channel for sleep intervention implementation, which could be further explored (e.g., to implement behavior control methods more systematically).

In the included reviews, the lack of behavioral theory in intervention development was surprising. Behavioral theory is used to effectively predict and alter many health behaviors, but its use has been very limited in sleep health [69,70]. In fact, only one included review specifically reported and emphasized the foundations of behavioral theory within its interventions [37]. While there is a small amount of research regarding the factors influencing sleep health and sleep behaviors [71,72], most interventions were not developed explicitly considering these factors, nor how they apply to specific target groups. More research into sleep-related factors and the application of theoretical frameworks of behavior change are lacking in the literature, requiring further research. To steer the reporting and comparability of reviews, following PRISMA guidelines is also heavily encouraged in future reviews.

## 5. Conclusions

This umbrella review is the first to provide an overview of strategies used in the rapidly evolving field of sleep health promotion, shedding light on target populations and intervention settings. Later school start times, behavior change methods, and mind–body exercise provided the most evidence of effectively improving sleep. Other interventions, such as sleep education, relaxation techniques, physical exercise, aromatherapy, massage, psychotherapy, and environmental interventions, also showed promising but inconsistent or limited results. Conclusions should be considered with caution, as there was high heterogeneity between studies. Nevertheless, this umbrella review can be seen as a first step towards reaching a greater understanding of sleep health promotion.

## Figures and Tables

**Figure 1 ijerph-18-05533-f001:**
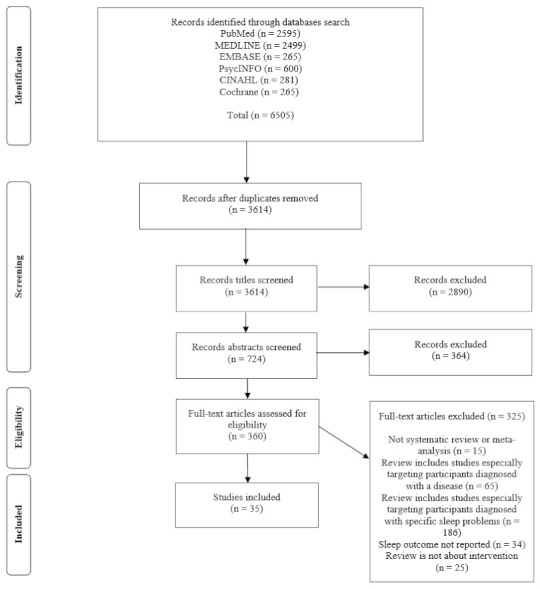
PRISMA diagram of the literature search.

**Table 1 ijerph-18-05533-t001:** Categories of intervention types, definition, and population targeted.

Intervention Type	Definition and Description	No. of Reviews [References]	Populations Targeted
Sleep Education	Delivering information about sleep in general and/or sleep hygiene tips in an instructional manner (for example in presentations, workshops, online modules, print media).	17 Reviews [29,30,32,33,34,35,36,37,38,39,40,41,42,43,44,45,46]	Infants, children, adolescents, college students, shift workers, athletes, adults
Behavior Change Methods (BCM)	Behavior change methods are based on learning theory, such as stimulus control, sleep schedules and structured routines; often managed by outside helpers such as parents, nurses, etc.	11 Reviews [29,32,33,34,38,40,42,45,47,48,49]	Elderly (in care facilities), infants, children, adolescents, hospitalized adults, adults, athletes, shift workers
Relaxation Techniques	Relaxation techniques such as mindfulness, breathing methods, meditation, guided imagery, progressive muscle relaxation, music, etc.	9 Reviews [31,35,41,44,45,48,50,51,52]	Adolescents, shift workers, college students, pregnant women, adults, hospitalized adults
Physical Exercise	Exercise, including Pilates, and other conditioning, or aerobic exercise; usually facilitated in group settings	7 Reviews [45,46,51,53,54,55,56]	Older adults (+60), postmenopausal women, pregnant women, postpartum women, middle-aged women, elderly (in care facilities)
Mind–Body Exercise (MBE)	Physical exercises with meditative components, including tai chi, yoga, Qigong, and the Rességuier Method	6 Reviews [31,51,53,54,55,57]	Older adults (+60), pregnant women, middle-aged women, adults, elderly
Aromatherapy and/or Massage	Aromatherapy is the therapeutic use of essential oils through inhalation or application on skin through massage; massage is the therapeutic manipulation of soft body tissues through kneading or rubbing; often massage and aromatherapy are implemented in combination	6 Reviews [46,48,50,51,56,58]	Hospitalized adults, pregnant women, adults, postpartum women, elderly (in care facilities)
Environmental	Changes to the physical environment including light therapy, noise reduction, or addressing any other sleep interruptions	5 Reviews [46,48,49,50,56]	Elderly (in care facilities), hospitalized adults, postpartum women, shift workers
Psychotherapy	Psychotherapy interventions administered online, in groups or individually, such as cognitive behavior therapy (CBT) and other psychotherapy including implosive therapy, cognitive refocusing treatment for insomnia, constructive worry, etc.	4 Reviews [40,41,44,46]	College students, shift workers, children, adolescents, postpartum women
Later School Start Times	Schools delaying start time by 20–85 min	4 Reviews [38,59,60,61]	Adolescents, children
Multicomponent	Combinations of two or more intervention types described above	18 Reviews [29,31,32,33,34,36,37,38,40,41,44,45,48,49,50,51,56,62]	Infants, children, adolescents, adults, college students, elderly (in care facilities), hospitalized patients, shift workers
Other	Other kinds of interventions, like dietary, hypnosis, biofeedback therapy, magnet therapy, drinking herbal tea, acupuncture, cryostimulation, and infrared light treatment	7 Reviews [31,42,44,46,50,51,63]	Elderly (in care facilities), postpartum women, adults, pregnant women, college students, hospitalized patients, athletes

**Table 2 ijerph-18-05533-t002:** Characteristics of and conclusions on the effectiveness of the intervention types of the included reviews: categorized by age category.

	Infants and Young Children					
Reference	Types and Descriptions of Interventions	Type of Review (No. of Eligible Studies Included in the Umbrella Review ^a^/Total No. of Studies in the Review)	Study Design	Population (Age Range)	Intervention Settings	Conclusions on Effectiveness
Bryanton et al. (2013) [29]	*Sleep Education* through print media, video, in-person and/or telephone meetings with a nurse Some interventions seem to combine both *Behavior Change Methods* with *Sleep Education*, but intervention descriptions are limited	Systematic review and meta-analysis (4/27)	RCTs	Infants and their parent	Varied: home, hospital, clinic setting	SD: Infant *sleep education* (often involving *Behavior Change Methods*) appeared to positively increase infant sleeping duration SQ: Impact on measures related to sleep quality were inconsistent SD: Three studies reported improvement in sleep duration at 6 weeks by more than an hour (MD = 62.08 min; 95% CI: 42.88 to 81.29; *p* < 0.00001)
Crichton and Symon (2016) [32]	*Sleep Education* through sessions, booklet, and phone calls with nurses *Behavior Change Methods*: self-settling, minimizing parental contact at night, etc. Some interventions seem to combine both *Behavior Change Methods* with *Sleep Education*, but intervention descriptions are limited	Systematic review (11/11)	RCTs, Quasi- experimental	Parents of infants (<6 months)	Varied: home, clinic, or hospital	SD/SQ: Studies of all included intervention types demonstrated significant improvements in sleep duration and/or night awakenings when techniques for self-settling, minimizing parental contact, and/or independent sleep cues were used (*p*-values and effect sizes not reported, but reported as significant)
Kempler et al. (2015) [33]	*Sleep Education* through leaflets, small group, booklets including sleep hygiene tips *Behavior Change Methods*: graded extinction, settling techniques, etc. *Multicomponent Interventions:* combining *Sleep Education* and *Behavior Change Methods*	Systematic review and meta-analysis (9/9)	RCTs	Infants (<12 months)	Varied; clinic, home	SD/SQ: Parent-directed interventions involving *Sleep Education* and/or *Behavior Change Methods* demonstrated some increases in parent-reported infant sleep duration, but no improvement in sleep quality, as measured by frequency of night awakenings SD: Small effect size observed for improving infant sleep duration by meta-analysis of 7 studies (SMD = 0.204 (Hedge’s *g*); 95% CI: 0.119 to 0.289; *p* < 0.01) when pooling all intervention types SQ: No evidence of improvement for infant night awakenings showed by meta-analysis of 6 studies (SMD = 0.103 (Hedge’s *g*); 95% CI: 0.032 to 0.238; *p* = 0.134), when pooling all intervention types
Mihelic et al. (2017) [62]	Interventions seem to combine *Sleep Education* and/or *Behavior Change Methods*	Systematic Review and Meta-analysis (13/36)	RCTs	Parents with infants (<12 months) or pregnant parents	Varied: home and hospital	SD/SQ: Combined *Sleep Education* and *Behavioral Change Methods* showed small increases in infant sleep duration and sleep quality, as measured by night awakenings (SMD = 0.24 (Cohen’s *d*); 95% CI: 0.14 to 0.35; *p* < 0.001)
Mindell et al. (2006) [34]	*Sleep Education* through sessions, printed materials, etc., including sleep hygiene tips *Behavior Change Methods*: scheduled bedtimes, reinforcement methods, standardized bedtime *Multicomponent Interventions*: combining *Sleep Education* and *Behavior Change Methods*	Systematic review (52/52)	RCTs, Quasi- experimental	Infants and children (<5 years)	Varied: home, clinic.	SQ: 94% of intervention studies involving *Sleep Education* and/or *Behavior Change Methods* were efficacious in improving sleep quality, as measured by frequency of bedtime problem and night awakenings (*p*-values and effect sizes not reported, but results stated as significant) SD: No significant effects on sleep duration found
**School-Aged Children and Adolescents**						
Arora and Taheri (2017) [35]	Primarily *Sleep Education* through group lessons *Relaxation Techniques*: mindfulness therapy	Systematic review (12/12)	NR	Adolescents (10–18 years)	Varied: school, research clinics	SD: Some *Sleep Education* interventions demonstrated some positive effects on sleep duration, but effects were inconsistent between studies There was insufficient data to draw a conclusion regarding the effectiveness of *Relaxation Techniques* (throughout the review, *p*-values and effect sizes not reported, but results stated as significant)
Aslund et al. (2018) [36]	Majority of the interventions were multicomponent, combining *Sleep Education*, *Psychotherapy*, *Behavior Change Methods*, and/or *Relaxation Techniques* One intervention may be limited to *Sleep Education*, but the description is limited	Systematic review and meta-analysis (6/6)	RCTs	Children and adolescents (6–20 years)	Varied: school, group therapy setting, internet, individual sessions	SD: Small increases in sleep duration showed by meta-analysis of 4 studies (MD = 11.47 min; SMD = 0.21 (Cohen’s *d*); 95% CI: 0.01 to 0.44; *p* = 0.05), through objective measures (actigraphy) SD: No significant effects observed on sleep duration at follow-up after 4–8 weeks SQ: Large decrease observed in objective SOL (MD = −19.48 min; SMD = −0.81 (Cohen’s *d*); 95% CI: −0.53 to −1.09; *p* < 0.0001) at post intervention SQ: Large decrease in objective SOL maintained at 4–8-week follow-up (MD = −23.67 min; SMD = −1.16 (Cohen’s *d*); 95% CI: −0.80 to −1.52 to; *p* < 0.0001)
Blunden et al. (2012) [37]	Exclusively *Sleep Education* interventions including print/online media, lectures, group-based lessons and/or workshops	Systematic review (12/12)	RCTs, Quasi- experimental	Children and adolescents (<20 years)	School	SD: Significant increases in sleep duration found in 2 of 12 studies (*p*-values and effect sizes not reported, but results stated as significant)
Busch et al. (2017) [38]	*Sleep Education* through lessons including sleep hygiene tips *Behavior Change Methods*: bedtime routines administered by parents *Later School Start Times* by 30 min *Multicomponent Interventions* combining *Sleep Education* on sleep hygiene tips *Physical Exercise*, and/or *Behavior Change Methods*	Systematic review (11/11)	RCT, Quasi- experimental	Children (4–12 years)	Varied: School, home, parent workplace, community settings	SD: Five of 11 studies displayed significant effects (increase of 8–45 min per night) on sleep duration SD: The five studies included 2 *Sleep Education* studies that showed significant effects on sleep duration after intervention, but not sustained at follow up, 2 *Multicomponent* studies showed significant effects on sleep duration, and it sustained from 6 months up to 2 years SD/SQ: Six of 11 interventions showed no significant effects on sleep duration, but showed reduction in SOL in 2 studies No conclusion for *Behavior Change Methods* and *Later School Start Times* because of limited evidence (*p*-values and effect sizes not reported, but results stated as significant)
Chung et al. (2017) [39]	*Sleep Education* through lessons, seminars, brochures, and leaflets including sleep hygiene tips	Systematic review and meta-analysis (7/7)	RCTs	Adolescents (10–19 years)	School	SD: School-based *Sleep Education* programs had short-term positive impacts on sleep duration, but benefits are not observed in follow-up SD: Small increases in weekday and weekend sleep duration observed immediately after the intervention (SMD = 0.23; 95% CI: 0.17 to 0.29; *p* = 0.0001 and SMD = 0.46; 95% CI: 0.04 to 0.88; *p* = 0.03)
Halal and Nunes (2014) [47]	*Behavior Change Methods* as taught to parents: positive routines, controlled comforting, gradual extinction, etc.	Systematic review (10/10)	NR	Children (<10 years)	Home/community	SQ: *Behavior Change Methods* demonstrated improvements in child sleep quality as measured by decreases in night awakenings, SOL, nocturnal awakenings (*p*-values and effect sizes not reported, but results reported as significant)
Marx et al. (2017) [61]	*Later School Start Times*	Systematic review and meta-analysis (8/11)	RCTs, crossover, Quasi- experimental	Secondary, middle, or high school- aged children (13–19 years)	School	SD: Meta-analysis of 2 studies showed a large increase in sleep duration for later school start times when compared to earlier start times (MD = 1.39 h; 95% CI: 0.38 to 2.39; *p* = 0.007) SD: 6 other studies not included in meta-analysis also demonstrated improvement in sleep duration
Meltzer et al. (2014) [40]	*Sleep Education* (via for example presentations, booklets) *Behavior Change Methods*: structured bedtime routine, bedtime pass, graduated extinction, scheduled awakenings *Psychotherapy*: CBT *Multicomponent Interventions*: combining *Sleep Education*, CBT, bright light therapy *(Environmental)*, and/or mindfulness *(Relaxation Techniques)*	Systematic review and meta-analysis (28/28)	RCTs, Quasi- experimental	Children (<18 years)	Varied: clinic, hospital, home	SQ: A small reduction in SOL in young children was demonstrated by a meta-analysis of 5 *Behavior Change Methods* interventions (SMD = −0.33; 95% CI: −0.48 to −0.18; *p* < 0.0001) SQ: Varied intervention types demonstrated a small reduction in the frequency of night awakenings in meta-analysis of 11 studies (SMD = −0.26; 95% CI: −0.35 to −0.17; *p* < 0.0001) No conclusion for *Psychotherapy* and *Multicomponent Interventions* because of insufficient data
Minges and Redeker (2016) [60]	*Later School Start Times* of 20–60 min later than control groups	Systematic review (6/6)	RCTs, Quasi experimental	Primary or secondary school-aged children (9–18 years)	School	Evidence supported *Later School Start Times* to improve adolescent sleep duration SD: Increases observed in sleep duration from 25 to 77 min per weeknight across studies
Morgenthaler et al. (2016) [59]	*Later School Start Times* of 20–85 min later than control groups	Systematic review and meta-analysis (18/18)	Quasi- experimental	High school-aged children (12–19 years)	School	SD: *Later School Start Times* were related to a longer weekday sleep duration, in particular when compared to 60 min earlier, found in meta-analysis of 2 studies (MD = 52.56 min; 95% CI: 38.74 to 66.37) SD: School-night sleep times increased in meta-analysis of 5 studies (MD = 18.65 min; 95% CI: 8.13 to 29.16) (throughout the review, *p*-values not reported, but results stated as significant)
**Adults**						
Barger et al. (2018) [41]	*Sleep Education* in different formats (presentations, workshops, online, literature) *Psychotherapy*: CBT *Relaxation Techniques*: mindfulness therapy *Multicomponent Interventions*: combining *Sleep Education*, *Physical Exercise*, stimulus control (*Behavior Change Methods*), and/or mindfulness (*Relaxation Techniques*)	Systematic review and meta-analysis (16/18)	RCTs, Quasi- experimental	Shift workers (>18 years)	Varied: home, work-based	SD/SQ: 9 of 16 studies showed improvement in sleep duration and/or quality (p-values and effect sizes not reported, but results stated as significant) SD/SQ: 7 of 16 studies showed mixed/inconclusive/or no effect in sleep duration and/or quality measures SQ: Large effects on global PSQI score at 4–8 weeks post baseline showed by meta-analysis of studies of 5 varying intervention types (SMD = 0.87 (Cohen’s *d*); 95% CI: 0.69 to 1.05; *p* < 0.00001) Results were combined across intervention types
Bonnar et al. (2018) [42]	*Sleep Education* through leaflets or lessons including sleep hygiene tips *Behavior Change Methods*: scheduled bedtimes with night bedtime extended/daytime naps, sleep-wake schedules, etc. *Other interventions*: red-light irradiation, cryostimulation	Systematic review (10/10)	Quasi- experimental, crossover	Athletes (>18 years)	Unclear	SD/SQ: *Sleep Education* consistently demonstrated positive effects on sleep duration and/or quality SD: *Behavior Change Methods* (scheduled bedtimes with night bedtime extended) consistently increased sleep duration, but daytime naps interventions showed no differences SD/SQ: Other interventions demonstrated improvement in sleep duration and/or quality, but with limited studies
Chung et al. (2017) [43]	*Sleep Education* through sessions, printed material including sleep hygiene tips	Systematic review and meta-analysis (15/15)	RCTs	Adults (NR)	Varied: Clinic or other healthcare setting	SD: Small increase in sleep duration showed in meta-analysis of 5 studies (MD = 25.06 min; 95% CI: 11.86 to 38.26; SMD = 0.28 (Hedge’s *g*); *p* < 0.001) SQ: Medium effect size on global PSQI scores showed in meta-analysis of 9 studies (MD = 1.75; 95% CI: 1.05 to 2.45; SMD = 0.51 (Hedge’s *g*); *p* < 0.001); SQ: Between group pooled analysis showed that *Sleep Education* is less effective in increasing sleep quality (PSQI) than the control groups (which received CBT and mindfulness)
Dietrich et al. (2016) [30]	*Sleep Education* through lessons and handouts	Systematic review (4/4)	RCTs, Quasi- experimental	College students (>18 years)	University	SQ: Insufficient evidence to determine effectiveness of *Sleep Education* or quality SQ: Three of four studies showed no difference (*p* > 0.05) and one showed significant improvement (*p* = 0.017) in global PSQI scores
Friedrich and Schlarb (2017) [44]	*Sleep Education* primarily focused on sleep hygiene, through lectures, discussions, online resources, and/or handouts *Relaxation Techniques*: progressive muscle relaxation, relaxing music, mindfulness, abdominal breathing, guided imagery, or insight meditation *Psychotherapy*: CBT, and other psychotherapeutic therapies, online or in-person *Multicomponent Interventions*: *Sleep Education, Relaxation Techniques, Psychotherapy*, and *Behavior Change Methods* (stimulus control) *Other Interventions*: hypnotherapy	Systematic review (27/27)	RCTs, quasi- experimental	College students (>18 years)	University	SD/SQ: *Sleep Education* interventions demonstrated no significant improvements in sleep duration but medium effects on varied measures of sleep quality (p-values and effect sizes not reported, but reported as significant) SQ: *Relaxation Techniques* showed medium effects on sleep quality SD/SQ: CBT *Psychotherapy* approaches resulted in large impacts on sleep duration, SOL, and other measures of sleep quality SD/SQ Non-CBT *Psychotherapy* interventions showed medium effects on sleep duration and some measures of sleep quality, but not SOL *Multicomponent Interventions* showed small effect on sleep quality (throughout the review, *p*-values and effect sizes not reported, but results stated as significant)
Hellström et al. (2011) [50]	*Relaxation Techniques*: music, music videos, guided muscle relaxation, mental imagery *Aromatherapy* and/or *Massage* *Environmental Interventions*: natural sounds *Multicomponent Interventions*: combining *Behavioral Change Methods*, such as routine management, and *Environmental Interventions*, such as adjusting light, temperature, and noise levels *Other Interventions*: acupuncture, magnetic pearl therapy	Systematic review (9/9)	RCTs	Adults (>19 years)	Healthcare setting (inpatient)	SQ: *Relaxation Techniques* had small to large effects on sleep quality. Music and music video showed large effects on sleep quality SQ: *Massage* had large effect size on sleep quality SD: *Multicomponent Interventions* yielded small effects on sleep duration No definite conclusion regarding *Environmental Interventions* and *Other Interventions* because of limited evidence (throughout the review, *p*-values and effect sizes not reported, but results stated as significant)
Hollenbach et al. (2013) [51]	*Physical Exercise*: aerobic exercise *Relaxation Techniques*: relaxation therapy *Mind–Body Exercise*: yoga *Massage* *Multicomponent Interventions*: *Physical Exercise* combined with *Sleep Education* *Other Interventions*: acupuncture	Systematic review (7/7)	RCTs, quasi- experimental	Pregnant women (15–45 years)	Varied; inpatient, and community	SQ: Non-pharmacological interventions to improve the sleep of pregnant women varied considerably and demonstrated little consistent evidence, but *Massage*, and to a lesser extent *Physical Exercise* and acupuncture were considered the most promising SQ: *Other Interventions* involving acupuncture showed improvement on varied measures of sleep quality (effect sizes not reported, but results often stated as significant *p* < 0.01)
Hwang and Shin (2015) [58]	*Aromatherapy* and/or *Massage*: inhalation aromatherapy, or massage combined with aromatherapy	Systematic review and meta-analysis (13/13)	RCTs, quasi- experimental	Adults (NR)	Varied; inpatient, home	SQ: Meta-analysis of 12 studies reported that *Aromatherapy Interventions* improved various measures of sleep quality (*p* < 0.001) SQ: Subgroup analysis showed that inhalation aromatherapy is more effective in improving sleep quality *p* < 0.001) than aromatherapy combined with massage therapy (*p* = 0.027) Only Z-scores reported, no SMD values for effect size
Knowlden et al. (2016) [63]	*Other Interventions:* Dietary interventions in real-life home (*in natura*) settings	Systematic review (4/21)	RCTs, Crossover	Adults (18–50 years)	Home	SD/SQ: No effect on sleep duration and/or quality observed in 4 studies
Murawski et al. (2018) [45]	*Sleep Education* through print media, online courses, and/or group-based classes, including sleep hygiene tips *Behavior Change Methods*: stimulus control, sleep schedules, etc. *Relaxation Techniques*: relaxation therapy, music, mindfulness, guided imagery *Physical Exercise*: aerobic exercise *Multicomponent Interventions*: combining *Behavior Change Methods* with mindfulness	Systematic review and meta-analysis (11/11)	RCTs, Quasi- experimental	Adults (18–64 years)	Varied: home, in-group, online	Intervention types were evaluated in combination SQ: Combination of varied intervention types showed a medium positive effect on the PSQI overall sleep quality measure in a meta-analysis of 9 studies (SMD = 0.52 (Hedge’s *g*); 95% CI: 0.24 to 0.80; *p* < 0.01) SD: A small positive effect on the PSQI sleep duration measure was shown in meta-analysis of 3 studies (SMD = 0.32 (Hedge’s *g*); 95% CI: 0.07 to 0.57; *p* = 0.01)
Neuendorf et al. (2015) [31]	*Relaxation Techniques*: progressive muscle relaxation, mindfulness, etc. *Mind–Body Exercise*: qigong studies, yoga studies, tai chi studies, Rességuier method *Multicomponent Interventions*: combining *Relaxation Techniques* with *Mind–Body Exercise* *Other Interventions*: biofeedback, hypnotherapy	Systematic review (112/112)	RCTs	Adults (>18 years)	Varied: home, hospital	SD/SQ: 50 studies that included *Relaxation Techniques* showed inconsistent effects on sleep duration and/or quality; inconclusive in 22 studies/positive effects in 19 studies/’mixed’ effects in 9 studies on sleep duration/quality SD/SQ: 29 studies that included *Mind–Body Exercise* showed inconsistent effects on sleep duration and/or quality; positive effects in 16 studies/inconclusive effects in 10 studies/’mixed’ effects in 3 studies on sleep duration/quality SD/SQ:11 studies that included *Multicomponent Interventions* showed inconsistent effects on sleep duration and/or quality; positive effects in 5 studies/inconclusive effects in 3 studies/’mixed’ effects in 3 studies on sleep duration/quality SD/SQ: 22 studies included *Other Interventions* including biofeedback and hypnotherapy showed inconsistent effects on sleep duration and/or quality; inconclusive effects in 12 studies/positive effects in 8 studies/mixed effects in 2 studies on sleep duration/quality (throughout review, *p*-values and effect sizes not reported, but results stated as significant)
Owais et al. (2018) [46]	*Sleep Education* through print media, individual meetings with a nurse, telephone support *Physical Exercise*: Pilates *Aromatherapy or Massage* (not combined); back massage, foot reflexology, use of essential oil necklaces, inhalation therapy *Environmental Interventions*: infant sleeping in mother’s room *Psychotherapy*: CBT *Other Interventions*: beverage (drinking hot teas); magnet therapy	Systematic review and meta-analysis (15/15)	RCTs, Quasi- experimental	Mothers between delivery and 12 months postpartum	Varied: home, clinic, and hospital	SQ: Interventions of multiple types showed medium-level improvements of subjective sleep quality in postpartum mothers, as shown in meta-analysis of 12 studies (SMD = 0.54 (Cohen’s *d*); 95% CI: 0.19 to 0.88; *p* = 0.00001) SQ: *Massage* demonstrated large effects on maternal subjective sleep quality in a meta-analysis of 3 studies (SMD = 1.07 (Cohen’s *d*); 95% CI = 0.79 to 1.34; *p* < 0.00001) SQ: Meta-analyses demonstrated no significant effects of *Aromatherapy*, *Sleep Education* and hot beverage (*Other*) interventions
Rubio et al. (2017) [53]	*Mind–Body Exercise*: yoga *Physical Exercise*: aerobic exercise	Systematic review and meta-analysis (5/5)	RCTs	Middle-aged women (48–56 years)	Community	SQ: *Mind–Body and Physical Exercise* improved PSQI global scores in middle-aged women as shown by a meta-analysis of 4 studies of both intervention types (MD = 1.34; 95% CI: 0.00 to 2.67; *p* = 0.05) SQ: Aerobic exercise improved global PSQI scores in a meta-analysis of 3 studies (MD = 1.85; 95% CI: 0.07 to 3.62; *p* = 0.04), while yoga alone showed no significant effects
Slanger et al. (2016) [49]	*Behavior Change Methods:* sleep schedules *Environmental Interventions*: bright light therapy *Multicomponent Interventions*: combining *Sleep Education* with *Physical Exercise*	Systematic review (12/17)	RCTs, Crossover	Shift workers	Workplace	SD: 2 studies examined sleep schedules but did not report sleep duration outcome data SD/SQ: 4 studies including bright light therapy demonstrated consistent but insignificant improvement in sleep duration and/or quality, while 2 studies did not report the results *Multicomponent Interventions* demonstrated improvement but not significantly while 2 studies did not report the sleep duration outcome data
Tamrat et al. (2013) [48]	*Behavior Change Methods*: stimulus control *Relaxation Techniques*: guided imagery audio, relaxing music videos *Aromatherapy* or *Massage* *Environmental Interventions*: daytime bright light, quiet time, white noise *Multicomponent Interventions*: combining *Relaxation Techniques* with *Massage* or *Environmental Interventions*	Systematic review (13/13)	RCTs, Quasi- experimental	Inpatient adults (NR)	Inpatient setting (hospital)	Some intervention types were not clearly described in terms of effectiveness such as *Relaxation Techniques, Massage, Aromatherapy, Behavior Change Methods* and *Multicomponent Interventions* SD/SQ: *Environmental Interventions* including daytime bright light and quite time improved sleep duration 7–18%, and white noise improved sleep quality by 38% in RCSQ scores (throughout review, p-values and effect sizes not reported, but results stated as significant)
**Elderly**						
Du et al. (2015) [54]	*Mind–Body Exercise*: tai chi *Physical Exercise*: shadow-boxing	Systematic review and meta-analysis (5/5)	RCTs	Older participants (>60 years)	Community settings	SD: *Mind–Body* and *Physical Exercise* were combined in a meta-analysis of two studies reporting medium effects on sleep duration (SMD = 0.55 (Cohen’s *d*); 95% CI: 0.21 to 0.9; *p* = 0.002) SQ: Tai chi (Mind body exercise) had positive effects on global PSQI scores in a meta-analysis of 5 studies (SMD = 0.87 (Cohen’s *d*); 95% CI: 0.49 to 1.25; *p* < 0.00001)
Koch et al. (2006) [56]	*Physical Exercise* *Aromatherapy and/or Massage* *Environmental Interventions*: noise reduction, reduction on number of night-time nursing care, routinized care, in structured daytime activity *Multicomponent Interventions*: Combinations of *Physical Exercise*, *Environmental Interventions, and Behavioral Change Methods*	Systematic review (41/41)	RCTs, quasi- experimental	Elderly (>65) years	Aged care facilities	Widely varied interventions demonstrated inconsistent improvements in sleep duration and/or sleep quality in elderly adults living in aged care facilities (throughout the review, results reported in narrative format)
Wu et al. (2015) [57]	*Mind–Body Exercise*: tai chi, yoga, qigong	Systematic review and meta-analysis (14/14)	RCTs	Elderly (>60 years) with complaints of poor sleep	Varied: senior centers/elderly homes, community center	SQ: *Mind–Body Exercise* had positive effect on global PSQI scores in older people with complaints of poor sleep in a meta-analysis of 12 studies (SMD = 0.70 (Cohen’s *d*); 95% CI: 0.43 to 0.96; *p* < 0.0001) SD: 2 studies demonstrated significant increases in sleep duration
Yang et al. (2012) [55]	*Physical Exercise*: supervised/group regular aerobic exercise or resistance training *Mind–Body Exercise*: one tai chi study	Systematic review (6/6)	RCTs	Older adults (>40 years)	Community	SQ: *Mind–Body and Physical Exercise* demonstrated positive effect size on global PSQI scores in meta-analysis that combined both types of interventions of 5 studies (SMD = 0.47; 95% CI: 0.08 to 0.86; *p* < 0.05) SD: There was no significant differences between groups
**Adults, Including the Elderly**						
De Niet et al. (2009) [52]	*Relaxation Techniques:* listening to calming music Music combined with additional *Relaxation Techniques*: progressive muscle relaxation, video, or relaxation instructions	Meta-analysis (5/5)	RCTs	Adults (>18 years) and elderly (>60 years)	Home or hospital setting, college	SQ: Music-assisted relaxation yielded medium positive effects on PSQI measures of sleep quality for patients with sleep complaints (SMD = 0.74 (Hedge’s *g*); 95% CI: 0.52 to 0.96; *p* < 0.0001) SQ: No significant differences in improvement in PSQI score between studies that used music solely and studies that combined music with relaxation (*p* = 0.45)

NR = Not Reported; RCTs = Randomized Controlled Trials; CBT = Cognitive Behavior Therapy; SMD = Standardized Mean Difference; CI = Confidence Interval; SD = Sleep Duration; SOL = Sleep Onset Latency; SQ = Sleep Quality; MD = Mean Differences; RCSQ = The Richards Campbell Sleep Questionnaire. ^a^ individual studies of reviews were not taken into account in this table (i.e., considered non-eligible) if these studies had no outcomes on sleep quality or sleep duration, or had irrelevant study designs for this umbrella review (e.g., correlational study).

## Data Availability

All relevant data are within the manuscript.

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
