# Peer review of "Sleep Health Promotion Interventions and Their Effectiveness: An Umbrella Review"

_ijerph, 2021, doi:10.3390/ijerph18115533_

Round 1

Reviewer 1 Report

I appreciate the opportunity to review this work, well thought out and explained in the article. I congratulate the authors. I believe that in its present wording it can be accepted. However I only make a small suggestion for your consideration: 1.- In the methodology indicate the full names indicated with acarnyms (for example: CINAHL or PICOS). 2.- In the categories of interventions, I suggest to group into the same category the methods of behavior change and those carried out with cognitive-behavioral therapy, because the two have similar theoretical principles and methodologies. And to remove from psychotherapy cognitive-condutual therapy leaving in this category the other therapies.

Reviewer 2 Report

Review of “Sleep Health Promotion Interventions and Their Effectiveness: 2 An umbrella review” (ijerph-1214589)

This umbrella review is focused on the effectiveness of sleep interventions in primarily healthy populations and showed that later school start times, behavior change methods, and mind-body exercise are relatively high possibility for improving sleep. This study is interesting. This reviewer had a one comment.

Please summarized the Table 2 according to the intervention type.

Reviewer 3 Report

Dear Authors and Editor,

The manuscript entitled "Sleep Health Promotion Interventions and Their Effectiveness: An umbrella review ". This is an umbrella review that determines what non-pharmacological sleep health interventions have been evaluated among healthy populations, by examining target groups, settings, and effectiveness in improving sleep quality and duration.

The manuscript is well developed and clearly responds to the stated objectives. The authors should make minor corrections.

1-Title:

  • Adequate: The authors include the study design in the title.

2-Abstract:

Change keywords. Remove the words ": sleep health", "sleep quality", "sleep duration", "sleep intervention", "sleep health promotion", and "Umbrella review". Not found in MeSH (Medical Subject Headings). 

3- Introduction:

  • adequate:The authors adequately summarize the background of the study.

4. Materials and Methods

PubMed is a free search engine accessing primarily the MEDLINE. Medline is the database.

  • adequate: The methodology is well developed. The authors have registered the review in PROSPERO. The authors have minimized bias.

5-Results

  • adequate:The flow chart is well developed and the topics are well identified. The authors have evaluated the selected articles.

6-Discussion

  • adequate: The authors correctly expose their strengths, limitations and practical implications.

 The authors only searched in English. This may be a limitation, even though most of the articles are in English.

7-References

Please revise the references as journal guidelines
